# CD4^+^ T Cell Responses to *Toxoplasma gondii* Are a Double-Edged Sword

**DOI:** 10.3390/vaccines11091485

**Published:** 2023-09-14

**Authors:** Kamal El Bissati, Paulette A. Krishack, Ying Zhou, Christopher R. Weber, Joseph Lykins, Dragana Jankovic, Karen L. Edelblum, Laura Fraczek, Harshita Grover, Aziz A. Chentoufi, Gurminder Singh, Catherine Reardon, J. P. Dubey, Steve Reed, Jeff Alexander, John Sidney, Alessandro Sette, Nilabh Shastri, Rima McLeod

**Affiliations:** 1Institute of Molecular Engineering, University of Chicago Medical Center, Chicago, IL 60637, USA; 2Department of Pathology, University of Chicago, Chicago, IL 60637, USA; pkrishack@gmail.com (P.A.K.); cweber@bsd.uchicago.edu (C.R.W.); gsingh@bwh.harvard.edu (G.S.); reardon@uchicago.edu (C.R.); 3Department of Ophthalmology and Visual Sciences, University of Chicago, Chicago, IL 60637, USA; yzhou@bsd.uchicago.edu (Y.Z.); joseph.lykins@bmc.org (J.L.); laura.fraczek@gmail.com (L.F.); rmcleod@bsd.uchicago.edu (R.M.); 4Department of Emergency Medicine, Chobanian & Avedisian School of Medicine, Boston University, Boston, MA 02215, USA; 5Immunoparasitology Unit, Laboratory of Parasitic Diseases, National Institute of Allergy and Infectious Diseases, National Institutes of Health, Bethesda, MD 20892, USA; djankovic@niaid.nih.gov; 6Department of Pathology, Molecular and Cell-Based Medicine, Icahn School of Medicine at Mount Sinai, New York, NY 10029, USA; karen.edelblum@mssm.edu; 7Center for Immunity and Inflammation, Laboratory Medicine, Department of Pathology, Rutgers New Jersey Medical School, Newark, NJ 07103, USA; 8Division of Immunology and Pathogenesis, Department of Molecular and Cellular Biology, University of California, Berkeley, CA 94720, USA; harshita.grover@slalom.com (H.G.); nshastr3@jhmi.edu (N.S.); 9Department of Medical Microbiology, Faculty of Health Sciences, University of Pretoria, Pretoria 0028, South Africa; achentou@uwo.ca; 10Animal Parasitic Diseases Laboratory, Beltsville Agricultural Research Center, Agricultural Research Service, U.S. Department of Agriculture, Beltsville, MD 20705, USA; 11Infectious Diseases Research Institute, 1616 Eastlake Ave E #400, Seattle, WA 98102, USA; steven.reed@hdt.bio; 12PaxVax, 3985-A Sorrento Valley Blvd, San Diego, CA 92121, USA; jalexander4469@outlook.com; 13La Jolla Institute of Allergy and Immunology, 9420 Athena Cir, La Jolla, CA 92037, USA; jsidney@lji.org (J.S.); alex@lji.org (A.S.)

**Keywords:** *Toxoplasma gondii*, HLA-A*11:01, vaccine, TLR4, CD4^+^ T cells

## Abstract

CD4^+^ T cells have been found to play critical roles in the control of both acute and chronic *Toxoplasma* infection. Previous studies identified a protective role for the *Toxoplasma* CD4^+^ T cell-eliciting peptide AS15 (AVEIHRPVPGTAPPS) in C57BL/6J mice. Herein, we found that immunizing mice with AS15 combined with GLA-SE, a TLR-4 agonist in emulsion adjuvant, can be either helpful in protecting male and female mice at early stages against Type I and Type II *Toxoplasma* parasites or harmful (lethal with intestinal, hepatic, and spleen pathology associated with a storm of IL6). Introducing the universal CD4^+^ T cell epitope PADRE abrogates the harmful phenotype of AS15. Our findings demonstrate quantitative and qualitative features of an effective *Toxoplasma*-specific CD4^+^ T cell response that should be considered in testing next-generation vaccines against toxoplasmosis. Our results also are cautionary that individual vaccine constituents can cause severe harm depending on the company they keep.

## 1. Introduction

*Toxoplasma gondii* infection is devastating to the host and causes damage to eyes or central nervous system (CNS), resulting in loss of sight, disability, and death. Immunocompromised individuals, including those with HIV/AIDS, and congenitally infected babies are more susceptible to developing disease [1,2,3,4]. Pyrimethamine and sulfadiazine are effective against actively replicating tachyzoites but cannot eliminate encysted bradyzoites. Thus, they cannot definitively cure the infection [5,6]. There is a great need for the development of a safe and potent vaccine to prevent the establishment of acute or chronic reactivating infection.

CD4^+^ and CD8^+^ T cells have a demonstrated role in protective immunity against *T. gondii* infection [7,8,9,10,11,12]. Vaccines that are composed of polypeptides that elicit epitope-restricted CD4^+^ and CD8^+^ T cells have the potential to induce an immune response against human toxoplasmic infection in humans. We are building a rational “immunosense” approach that utilizes epitope selection for peptides that can be presented by human major histocompatibility complexes (MHCs) to elicit protective CD8^+^ and CD4^+^ T cells [13,14,15,16]. We have made considerable efforts to identify different CD8 epitopes derived from different antigens of the *Toxoplasma* life cycle that can be presented by different HLAs of MHC class I in different vaccine platforms [14,15,16]. Our vaccine formulations have the universal CD4^+^ helper T cell epitope PADRE and GLA-SE adjuvant [14,15,16]. PADRE has been shown to play an important role in the priming of CD4^+^ T cells that assist the generation of memory T cell populations [17]. In vitro in human PBMCs and in vivo in mice, PADRE contributes to the generation of responsive CD4^+^ T cells in addition to assisting the generation of antigen-specific CD8^+^ T cells in vaccination strategies [18].

Grover et al. discovered a CD4^+^ T cell-stimulating 15-mer *T. gondii* peptide (AS15) from a putative transmembrane protein of *Toxoplasma gondii* and showed that immunization with this peptide prior to infection leads to lower parasite burden in the brain of C57BL/6 strain mice challenged with Me49 strain parasites [19]. CD4^+^, not CD8^+^, T cells are the primary populations responsible for interferon-gamma (IFN-γ) production in splenocytes from these mice immunized with AS15 [19]. Thus, a protective CD4^+^ T cell response to *T. gondii* plays an important role in the design of more effective vaccines.

In the current study, we aimed to gain insights into the inclusion of AS15 and enhancement of *T. gondii*-specific CD8^+^ T cell immune responses by this peptide in vaccinated mice. To address this, we characterized the effect of AS15 peptide immunization in conjunction with the adjuvant GLA-SE (lipid TLR-4 agonist) in HLA-A*11:01 transgenic mice. This was noted to elicit robust increases in the levels of circulating cytokines, as well as to enhance survival after challenge with both virulent Type I and Type II *T. gondii*. When combined into a single formulation with previously identified CD8^+^ T cell-stimulating peptides and the universal CD4^+^ T cell-stimulating epitope PADRE, reductions in the production of IFN-γ were observed, despite conferral of similar protection. It was also determined that stimulation of CD4^+^ T cells provided protection against lethal infection by *T. gondii.* Vaccine formulations containing CD8^+^ T cell-stimulating epitopes, on the other hand, afforded protection to mice challenged upwards of 30 days after immunization. This protection was accompanied by statistically significant increases in circulating memory T cells in this latter group. Surprisingly, boost immunization with adjuvanted AS15 alone resulted in significant intestinal pathology and death in a subset of female mice, apparently caused by cytokine storm. The inclusion of CD8^+^-stimulating epitopes and PADRE rescued this phenotype, and no deaths were observed after boost immunization in these mice.

## 2. Materials and Methods

### 2.1. Peptides and Adjuvants

The peptides described in this study were synthesized by Synthetic Biomolecules, San Diego, CA, USA at >90% purity in a lyophilized form [13]. These include the following: the CD4^+^ T cell-stimulating peptide AS15 (AVEIHRPVPGTAPPS), the universal human epitope PADRE (AKFVAAWTLKAAA), and the CD8^+^ T cell-stimulating peptides (GRA6_164–172_ (AMLTAFFLR), SAG1_224–232_ (KSFKDILPK), SAG2C_13–21_ (STFWPCLLR), SRS52A_250–258_ (SSAYVFSVK), GRA5_89–98_ (AVVSLLRLLK)). The adjuvant used in this study is GLA-SE, a TLR4 agonist, synthesized by the Infectious Diseases Research Institute (IDRI, Seattle, WA, USA), as previously described [20].

### 2.2. Mice

Female and male HLA-A*11:01 transgenic mice used in these experiments, expressing chimeric genes HLA-A*11:01 and H-2K^b^, were created in a C57BL/6 background. More details have been reported previously [13]. The mice studies were performed at the University of Chicago and USDA facilities at Beltsville Agricultural Research Center, Animal Parasitic Diseases Laboratory with the approval of the Institutional Animal Care and Use Committee at the University of Chicago (protocol# 71734).

### 2.3. Immunizations of Mice

HLA-A*11:01 mice were inoculated subcutaneously at the base of the tail using a 30-gauge needle with 50 μg of peptides emulsified in 20 μg of GLA-SE 3 times at 2-week intervals. For challenge studies, we infected HLA-A*11:01 mice intraperitoneally with either 2000 RH-YFP tachyzoites (Type I) or 2000 ME49-Fluc (Type II) parasites 14 days after the last immunization. Control mice were injected with either PBS or 2000 tachyzoites of the live attenuated mutant strain of the ribosomal proteins S13 gene (*rps13Δ*), as an internal control.

### 2.4. Flow Cytometry

Standard flow cytometry was used as described in [16] for testing IFN-γ expression and CD8^+^/CD4^+^ T cell memory response. The following conjugated antibodies from eBiosciences (San Diego, CA, USA) were used: CD3-APC (145-2C11), CD4-PE (GK1.5), CD8-PerCP (53-6.7), CD44-AF780 (IM7), CD127-FITC, and CD44^hi^CD127^+^. After washing, splenocytes were fixed before being measured on a BD LSR II flow cytometer (BD Biosciences) and analyzed using FlowJo software 10.0 (Tree Star, Ashland, OR, USA). The MHC class II tetramers, which present AS15 to T cells, were obtained from the NIH tetramer facility and used as described previously.

### 2.5. ELISpot Assays

In order to assay for production of IFN-γ by murine splenocytes, ELISpot assays were performed. Mice were euthanized 7 to 14 days after the last immunization. Splenocytes were processed as described previously [14]. The peptide concentration used for stimulation of splenocytes was 20 µg/mL. ELISpot assays with murine splenocytes were performed using α-mouse IFN-γ mAb (AN18) and biotinylated α-mouse IFN-γ mAb (R4-6A2) as the cytokine-specific capture antibodies. Antibodies were monoclonal antibodies.

### 2.6. T Cell Cytotoxicity Assay

In order to detect cytolytic CD8^+^ T cells recognizing AS15 peptide in freshly isolated spleen cells, we used the CD107a/b cytotoxicity assay as described in [21].

### 2.7. Bioluminescence Imaging of Mouse Brains to Determine Cyst Burden

Ex vivo mouse brains were analyzed for the presence of cysts using an imaging system as described previously. Briefly, 200 mL (15.4 mg/mL) of D-luciferin (GoldBio, Olivette, MO, USA) was injected retro-orbitally into mice. After 10 min, mouse brains were collected. Bioluminescence imaging was performed at the University of Chicago Integrated Small Animal Imaging Research Resource on an IVIS Spectrum (PerkinElmer, Waltham, MA, USA). Imaging analysis was performed using Living Image 4.7.3.

### 2.8. Quantification of Cysts in Mouse Brains after Challenge

Next, 21 days after the challenge, brains were homogenized in saline, and cysts were quantified using an optical microscope. The number of cysts was confirmed by staining brain cysts with fluorescein-labeled Dolichos biflorus agglutinin (Vector Laboratories) and quantifying using a fluorescence microscope.

### 2.9. Hematoxylin and Eosin (H&E) and Histochemical Staining Analysis

Mouse intestines and livers were fixed in paraformaldehyde, embedded with paraffin, and then cut into 4 µm sections for H&E staining. Sections were stained with rabbit anti-CD3 (Abcam) or rabbit caspase 3 (Abcam).

### 2.10. Statistical Analyses

Data comparisons between groups were made using analysis of variance (ANOVA) or a Student’s *t* test [6]. We used GraphPad Prism 10 software (GraphPad Software, San Diego, CA, USA) for comparison. The results are shown as the means of 2 independent experiments (*n* = 10 mice) ± standard deviation. Significant differences relative to untreated control were determined using Student’s *t* test. The results were considered to be statistically significant at *p* < 0.05.

## 3. Results

CD4^+^ T cell-stimulating peptide AS15 with the adjuvant GLA-SE (lipid TLR-4 agonist) resulted in increased IFN-γ production.

Splenocytes were isolated from immunized HLA-A*11:01 transgenic mice 24 h after the second immunization. Their ability to generate IFN-γ and lymphocyte proliferation in response to AS15 peptide was assessed. The data in Figure 1, showing IFN-γ secretion in ELISpot, indicate that IFN-γ secretion induced by AS15 peptide stimulation was significantly enhanced when mice were previously immunized with adjuvanted AS15 peptide and GLA-SE adjuvant, but not with AS15 peptide alone or the adjuvant alone (*p* < 0.01).

Addition of HLA A-11-restricted CD8^+^ T cell epitope peptides and/or PADRE modulated IFN-γ expression.

Since IFN-γ and CD8^+^ T cells have been shown to be critical for the survival of mice after infection with *Toxoplasma*, we combined AS15 peptide, PADRE, and HLA-A*11:01-restricted CD8^+^ peptides to attempt to enhance the production of IFN-γ and better protect against *T. gondii*. Immunization with AS15 and GLA-SE (AS15-GLA-SE) induced IFN-γ production from CD4^+^ T cells (Figure 2A). Surprisingly, when the universal helper T cell epitope PADRE was added to AS15 plus GLA-SE, the amount of IFN-γ produced was significantly reduced (*p* < 0.001). When HLA-A*11:01-restricted CD8^+^ T cell epitopes from GRA6_164–172_ (AMLTAFFLR), SAG1_224–232_ (KSFKDILPK), SAG2C_13–21_ (STFWPCLLR), SRS52A_250–258_ (SSAYVFSVK), and GRA5_89–98_ (AVVSLLRLLK) were added [16], robust CD8^+^ T cell production of IFN-γ was observed (Figure 2B).

### 3.1. Increased Levels of CD4^+^ AS15 Tetramer-Specific Cells Indicate T Cell Responses Are Specific to AS15

MHC class II I-A^b^–AS15 tetramer staining was performed with splenocytes from immunized and control PBS mice. High percentages of tetramer-positive cells in female mice immunized with AS15 alone, AS15 + GLA-SE, or AS15 + GLA-SE + PADRE in comparison with the groups that did not receive AS15 peptide are shown in Figure 2C. Increased levels of CD4^+^ tetramer-specific T cells following stimulation indicate the specificity of T cell response to AS15. When CD8^+^ T cell-eliciting peptide was added in the immunogens, the percentage of tetramer-positive CD4^+^ T cells diminished. The mean of tetramer-positive CD4^+^ T cells also significantly diminished when PADRE was added (*p* < 0.05), suggesting changes in the distribution of CD4^+^ T cells when other immunogens are added.

### 3.2. Boosted Immunization with AS15 and GLA-SE Resulted in Death in Immunized Female HLA-A11 Transgenic Mice, but Not Males

To determine whether increased IFN-γ expression would result in increased survival for infected mice following immunization, both male and female transgenic mice were immunized with a prime and two weeks later with a boost immunization. Surprisingly, half of the female mice demonstrated marked sensitivity to boosting with AS15 plus GLA-SE. At day 3–4 post-boost immunization, 62% of these mice died (Figure 3A). No deaths were observed in male mice (Figure 3B). Because the finding was surprising and reproducible, we investigated this finding in more detail by characterizing the response of mice to AS15-GLA-SE. We did not include this group in our subsequent survival analyses. The addition of PADRE alone or in combination with the CD8^+^ peptide pool in the immunogen eliminated the increase in mortality (Figure 3C,D), which paralleled the decrease in IFN-γ-producing T cells and the competitive effect of the other peptides in reducing tetramer-positive CD4^+^ T cells. These findings were indicative that the number of AS15-responsive T cells is likely to be responsible for the enhanced mortality observed in female mice.

### 3.3. Peptide That Elicits Specific CD4^+^ T Cells Protects after Immunization against Challenge with Me49 Parasites at Early but Not at Late Stages

Next, the ability of AS15-GLA-SE-PADRE and AS15-CD8^+^ T cell-restricted peptides-PADRE-GLA-SE to protect mice against Me49 Type II *T. gondii* was evaluated. Female mice were immunized and subsequently challenged after 10 or 35 days with 2000 Me49 (Fluc) that expressed the Firefly luciferase (FLUC) gene. The brains of these mice were imaged at 21 days post-challenge using an IVIS imaging system. As shown in Figure 4A, the stimulation of CD4^+^ T cells by AS15 + GLA-SE + PADRE contributed to enhanced survival of mice infected with *Toxoplasma* 10 days but not 35 days post-immunization (*p* < 0.001). In contrast, when HLA-A*11:01-restricted CD8^+^ T cell-eliciting peptides were present in immunizations of mice, with or without AS15, protection was conferred in groups challenged 35 days post-immunization (*p* < 0.001) (Figure 4B). This was confirmed through reductions in the number of luciferase-expressing parasites in the brain (Figure 4C,D). At 21 days post-challenge, both AS15 and the CD8 peptide pool plus AS15 (both containing PADRE) were effective in reducing brain parasite burden. Parasites attenuated through deletion of the ribosomal protein S13 gene (*rps13Δ*) [22] were used for immunization as a positive control (Figure 4C). Taken together, this shows that AS15 peptide immunization conferred protection only when the challenge occurred very close to the time of immunization (10 days), but provided no longer-term protection (35 days). Only CD8-stimulating peptide pools provided long-term protection, and the addition of AS15 was not helpful.

### 3.4. The Addition of CD8^+^ T Cell-Eliciting Peptides plus GLA-SE to PADRE Increases Memory CD8^+^ T Cell Response

We then analyzed the effect of PADRE or AS15 in combination with CD8^+^ T cell peptides on the number of *T. gondii*-specific memory T cells. When spleens from mice immunized with CD8^+^ T cell-eliciting peptides in combination with GLA-SE and PADRE were analyzed 35 days after immunization, CD8^+^ memory T cells were significantly increased for mice immunized with CD8^+^ T cells peptides plus PADRE plus GLA-SE when compared with the GLA-SE adjuvant alone or PBS (Figure 5C,D). No significant increase in CD4^+^ T cell memory response was observed in any group (*p* > 0.05) (Figure 5A,B).

### 3.5. AS15 plus GLA-SE Protects Mice against Virulent Type I Toxoplasma gondii

We immunized HLA-A*11:01 mice twice with AS15-GLA-SE, AS15-GLA-SE-PADRE, CD8^+^ T cell-eliciting HLA-A*11:01-restricted peptides-PADRE-GLA-SE, or AS15-CD8^+^ T cell-eliciting HLA-A*11:01-restricted peptides plus PADRE. PBS or GLA-SE adjuvant alone were used as controls. We challenged mice 10 days post-immunization with *T. gondii*, Type I, RH strain. Peritoneal fluid was collected 5 days post-infection, and parasite fluorescence and numbers were measured using a fluorometer or hematocytometer, respectively. Compared to the control, the number of fluorescent parasites from all immunized mice was significantly lower (*p* < 0.05) (Figure 6A). This reduction was also observed in measurements of total parasite burden. A subset of female mice immunized with AS15-GLA-SE (*n* = 5 mice) exhibited significant weight loss (>40% total body weight) and died within 3 days after the second immunization. The mice that did survive (*n* = 3 mice) showed the most robust protection against RH tachyzoites (Figure 6A). Interestingly, no clinical symptoms were observed in similarly immunized male mice following the boost immunization. None of these male mice (*n* = 15) died, and all were resistant to subsequent infection with Type I parasites (*p* < 0.0001) (Figure 6B).

### 3.6. AS15 with the Adjuvant GLA-SE Increases Cytokines, Correlating with Significant Pathology and Death in HLA-A11 Transgenic Mice

To determine how the immunogen-adjuvanted AS15 was affecting the fate of mice, and to understand how we might subsequently be able to avoid the loss of mice after the second immunization, we measured the serum levels of IFN-γ, IL-12p70, IL-2, IL-10, and IL-6 using a cytokine bead array. Serum cytokines were measured at 24 h before and after the second immunization (Figure 7). When AS15 was combined with GLA-SE, there was a significant increase in the level of IFN-γ and IL-6 24 h post-boost immunization compared with sera obtained before the second immunization. When mice were immunized with the universal helper T cell epitope PADRE in combination with AS15 and the adjuvant GLA-SE, the immune response was modulated such that there were lower levels of cytokines, including IFN-γ and IL-6, produced (Figure 7).

We also evaluated whether weight loss and mortality in these mice were associated with small bowel and liver damage during infection (Figure 8A,B). Histological analysis of H&E-stained small intestinal sections showed numerous apoptotic bodies within small intestinal crypts in mice treated with AS15 + GLA-SE (white square). Apoptosis was confirmed through immunohistochemical staining for cleaved caspase 3 (Figure 8A). This was also associated with increased infiltration of CD3^+^ T cells within the ileal epithelium (Figure 8A). In contrast, the ileum of untreated mice appeared normal, with long and slender villi, no increase in apoptosis, and only occasional intraepithelial CD3^+^ lymphocytes. Changes also occurred in the liver of treated mice (Figure 8B), which consisted of microscopic necrotic foci with acidophil bodies and neutrophils surrounded by CD3^+^ T cells. The livers of untreated mice appeared normal, without necrotic foci or increased inflammatory cell infiltrate. The addition of PADRE abrogated this pathology. There was no significant necrosis or apoptosis observed in the tissues of these mice, in stark contrast to the female mice immunized with AS15-GLA-SE.

### 3.7. No Difference between Vaccinations for the Bulk Frequencies of FOXP3^+^ CD4^+^ T Cells and CD107a and -b Staining

To determine why PADRE abrogates the severe pathological changes triggered in a subset of mice by immunization with AS15 and GLA-SE, we examined the expression of FoxP3, a marker for CD4^+^CD25^+^ T regulatory cells, and CD107a and -b, markers of cytotoxic T cells. We found that there was no difference in the expression of FoxP3 between the two groups, suggesting that the effect of PADRE is not mediated by the activation of CD4^+^CD25^+^ T regulatory cells (Figure 9A). Additionally, there was no difference in the saining of CD107a/b^+^ T cells, although some female mice from the group stimulated with AS15 had high staining (Figure 9B).

## 4. Discussion

To date, there is no vaccine for toxoplasmosis. Live attenuated *T. gondii* tachyzoites induce a protective cell-mediated immune response in animals but have been considered unsafe for humans [22,23]. Consequently, investigation into vaccine development against toxoplasmosis has focused on the use of subcomponent vaccines such as full-length proteins [24,25,26,27], DNA [28,29,30,31], or antigenic peptides [32,33] that trigger host immune response. CD8^+^ T cells are essential for the control of *T. gondii* by increasing levels of cytokines and their cytolytic activities [8,34,35]. IFN-γ and IL-12 are important for the protection of mice against *Toxoplasma* [36,37,38]. IL-6 was shown to play a protective role against the development of toxoplasmic encephalitis by stimulating IFN-γ production [39].

We have made considerable efforts to identify promising vaccine candidate antigens for *T. gondii* of potential utility in humans by developing a logical immunosense approach [13,14,16]. Our previous work reported many protective epitopes recognized by T cells of immune persons in a human leukocyte antigen (HLA). These were confirmed in our HLA transgenic mice model. We identified multiple HLA Class I-restricted epitopes including those restricted for HLA-A*02:01, HLA-A*11:01, HLA-B*07:02 and others pertinent to the present work [13,40]. We also demonstrated a cross-presentation process of HLA class I and antigen processing in the proteasome [16]. We selected five HLA-A*11:01-restricted epitopes from antigens expressed during tachyzoite and bradyzoite parasite life cycle stages to develop a vaccine [13,40]. Our five pooled HLA-A*11:01 peptides combined with a universal helper T epitope, PADRE, elicit IFN-γ from human HLA-A*11:01 supertype-restricted CD8^+^ T cells, and adjuvanting with GLA-SE increases memory T cells and reduces cysts in the brains of HLA-A*11:01 transgenic mice [16].

We also defined a novel delivery platform to promote strong CD4^+^ and CD8^+^ T cell responses. We designed and produced a single epitope string containing five HLA-A*11:01-restricted epitopes and PADRE with different spacers and found an increase in memory T cells and reduction in cysts in the brains of HLA-A*11:01 transgenic mice [15,16]. Self-Assembling Protein Nanoparticles (SAPNs), another delivery platform we tested, also induced a very strong immune response due to a repetitive display of antigens [41,42,43]. We constructed nanoparticles delivering a single B07-restricted CD8^+^ T cell epitope from dense granule protein (GRA7_20–28_) in conjunction with PADRE and found a substantial reduction in cyst burden in the brains [14].

The efficacy of a vaccine also depends on the immune-stimulatory properties of adjuvants (such as the lipid TLR-4 agonist GLA-SE) and a delivery system for both the antigen and the adjuvant. We chose GLA-SE because we have demonstrated in our previous studies that GLA-SE was superior to ALUM as an adjuvant for our polypeptide vaccine against *Toxoplasma* [16]. GLA-SE is being investigated in preclinical or clinical studies for *Leishmania*, cancer, and tuberculosis vaccines [15,44,45]. Vaccination with GLA-SE and the recombinant fatty acid binding protein Sm14 enhances humoral and cell-mediated immunity and protects humans and animals against schistosomiasis [46].

In addition, we found that specific *T. gondii* CD4^+^ T cell-eliciting epitopes are needed as components for vaccine protection. Although PADRE is critical to our vaccine formulation to generate IL-2 to expand CD8^+^ T cells, it is a promiscuous and universal CD4^+^ T cell epitope and not specific for *T. gondii*. Therefore, for memory responses to a pathogen so far there, was no *T. gondii*-specific CD4^+^ T cell help. Additional CD4^+^ T cell epitopes from immunogenic proteins expressed from *T. gondii* life cycle stages are needed. Herein, we wanted to determine whether adding a *T. gondii*-specific peptide that elicits mouse CD4^+^ T cells would enhance protection. Previous studies have shown that the CD4^+^ T cell-stimulating peptide AS15 confers significant protection against toxoplasmosis in C57BL/6 mice [19]. In this study, we compared the effectiveness of adjuvanted AS15 in combination with PADRE and CD8^+^-restricted peptides. These five CD8^+^ T cell-restricted peptides bind to the HLA-A03 supertype. We compared the protective immune responses generated against toxoplasmosis in HLA-A*11:01 transgenic mice. Previous studies found that the precursor frequency of T cells responding to this peptide was unusually high and that there was remarkably low reactivity to any cross-reactive T cell epitopes [47]. The binding of the mouse allele H2-IAb is 3.29 nM.

Nonetheless, these documented benefits of the stimulation of Th1 responses, however, have a significant potential limitation, as observed in our female mice immunized with AS15 and GLA-SE. Our findings show AS15 with GLA-SE induces high levels of cytokines in a subset of immunized mice, with elevated levels of IFN-γ and IL-6, which are associated with severe intestinal and liver pathology. Those mice that survived the process of induction of a high level of cytokines and subsequent necrosis were highly resistant to Type I and II *Toxoplasma* strains. These observations occur when GLA-SE and AS15 are present. GLA is a TLR4 ligand that leads to the activation of myeloid differentiation primary response 88 (MyD88) and downstream signaling pathways [44,48]. GLA-SE entraps proteins and peptides in nanoparticle structures, although the precise location of AS15 peptide was not studied herein. Our data corroborate previous studies showing that IFN-γ-mediated pathology in the small intestine could lead to death in C57BL/6 mice [49]. In these earlier studies of IFN-γ toxicity, treatment with anti-IFN-γ mAb after mice developed clinical illness helped significantly increase survival [49]. In conclusion, our study suggests that IFN-γ and IL-6 could be a double-edged sword. They can contribute to protection but, in excess, can thereafter contribute to the death of mice.

Previous studies have shown that immunization of C57BL/6 mice with AS15 elicited a potent CD4^+^ response, but it is not a CD8^+^ T cell-eliciting epitope [19]. For this reason, we sought to combine AS15 peptide, PADRE, and HLA A-11-restricted CD8^+^ peptides to enhance IFN-γ production and improve protection against *T. gondii* through increased stimulation of CD8^+^ T cells. The addition of PADRE, which elicits another restricted CD4^+^ T cell, to AS15 peptide reduces the increase in IFN-γ from CD4^+^ T cells and abrogates the AS15 phenotype and death.

We did not observe the presence of apoptotic cells or necrosis in the liver and small intestine when PADRE was added to the vaccine formulation. Furthermore, AS15 plus PADRE seem to play a protective role immediately after booster immunization. In contrast, CD8^+^ T cell-eliciting peptides help to protect mice after the immunization period. This is likely explained by the long-lasting immune response and contribution of CD8^+^ T cell-restricted peptides to an increase in the number of circulating memory CD8^+^ T cells.

It is not clear why PADRE abrogates the severe pathological changes triggered in a subset of mice by immunization with AS15 and GLA-SE. One hypothesis that might explain this finding could be that PADRE might activate T regulatory cells (Treg) to reduce the response induced by AS15. To test this hypothesis, we examined the PADRE- and AS15-specific CD4^+^ T cells for their expression of FoxP3, a marker for CD4^+^CD25^+^ T regulatory cells. We found that there was no difference in the expression of FoxP3 between the two groups, suggesting that the effect of PADRE is not mediated by the activation of CD4^+^CD25^+^ T regulatory cells (Figure 9A).

Studies with tetramers recognizing AS15-responsive CD4^+^ T cells (Figure 2C) demonstrate that when PADRE is included in the immunogen along with GLASE and AS15, there is a marked and significant decrease in AS15-responsive CD4^+^ T cells. This results in protection from the otherwise lethal IFN-γ and IL-6 cytokine storm and harmful pathology in the intestines associated with these CD4^+^ T cells. This indicates the likelihood of a pool of precursor cells that respond to PADRE and to AS15 that reduces the pathogenic response. Since there is no difference in regulatory T cells under these conditions, this suggests that it is simply a decrease in these strongly responsive T cells. As neither AS15 nor GLA-SE alone elicit this harmful response but only GLA-SE plus AS15, the emulsion with the TLR4 ligand is essential for eliciting these very reactive T cells, which, when present, can be lethal (Appendix A).

This is reminiscent of the work of Nelson et al. [47]. Their study noted that T cell receptor (TCR) cross-reactivity between MHC Class II-binding self and foreign peptides appeared to influence naive CD4^+^ T cell repertoire and autoimmunity. They found that nonamer peptides that bind to the same MHCII molecule only need to share five amino acids to cross-react on the same TCR [47]. Their findings also suggest that many TCRs focus primarily on the TCR contact amino acids no matter how the peptide is anchored to MHCII. This conclusion is supported by the previous work of Birnbaum using structural biology studies [50]. Of interest, although there are not five identical amino acids in PADRE and AS15, seven of the eight N terminal amino acids and also five of the C terminal amino acids are homologous in PADRE and AS15 (e.g., **AKFVA**A**WT**LKA**A**A and **AVEIH**R**PV**PGT**A**P, with bold showing the conserved (not identical) residues). This overlap suggests they might bind TCRs sufficiently to result in the reduction in the precursor pool and cause the effect we note.

Alexander et al. [51] found previously that PADRE bound with relatively high affinity to I-A^b^ [52]. To determine if the toxicity is alleviated simply because the IAb peptide binding of PADRE is greater than that of AS15, we performed a new competitive binding study of AS15 for I-A^b^ (IC50~44nM) using the procedure described by [51]. We found that AS15 bound weakly to I-A^b^ (413nM). Our data effectively confirm the lower frequencies of AS15 tetramer-positive CD4 T cells after PADRE + AS15 + GLASE compared to AS15 + GLASE, which is what can be observed in Figure 2C.

We also examined the T cell stimulation generated by AS15 or AS15 + PADRE for their expression of CD107a and -b, markers of cytotoxic T cells. We found that some female mice from the group stimulated with AS15 demonstrated high levels of CD107a/b^+^ T cells (Figure 9B). This correlates with the variation observed in outcomes following boosted immunization.

Another hypothesis that the success of PADRE in enhancing antigen-specific CD4^+^ T cell immune responses and balancing the *Toxoplasma* CD4^+^-restricted AS15 response warrants further exploration of innovative strategies that are capable of generating CD4^+^ T cell immune responses.

Since the ultimate objective of our work is to develop a vaccine for humans, it will be important for AS15 to bind efficiently to a human MHC Class II, DRB01:01, as a proof-of-principle approach in the future. Although the coverage of this allele is not widely represented worldwide, it provides further conceptual support for the use of a *Toxoplasma* CD4^+^ T cell-eliciting epitope for our vaccine formulation. Recent studies regarding the processing of *T. gondii* in human cells have found longer peptides (likely decoy) that bind to HLA-A2 during infection of THP1 cells with *Toxoplasma* [53]. Therefore, our immunosense approach for work to identify the protective candidate epitope in silico and in vitro in binding assays with human MHC class cells and testing them in HLA transgenic mouse models represents a rational approach for the development of *T. gondii* vaccines that could produce an armamentarium of protective CD8^+^ T cells before decoy peptides are generated.

Taken all together, it should be useful to combine into a single vaccine formulation of epitopes that stimulate both *T. gondii* CD4^+^ and CD8^+^ T cells effective against *T. gondii,* as well as appropriate adjuvants and universal epitopes important for initiating immune responses. This combination is designed to confer protection soon after a boost immunization via CD4^+^ stimulation as well as enhanced protection over the longer term via the production of memory T cells through the inclusion of CD8^+^-stimulating epitopes. This balanced approach to vaccine development seems the most likely to confer robust protection against toxoplasmosis while avoiding serious immunological complications, preventing morbidity and mortality.

## 5. Conclusions

Our study touches on a very important issue regarding the balance between the protective versus pathogenic effects of a vaccine against human toxoplasmosis. Using human HLA-11:01 transgenic mice, the universal HLA-DR and mouse IA-b-binding PADRE epitope, adjuvanted with GLA-SE, as well as previous knowledge regarding immuno-stimulating parasite CD4 and CD8 epitopes in mice, we demonstrate specific combinations that promote mouse survival after challenge. Our quest to find the best combination of immunogens reveals the importance of CD4^+^ to confer protection soon after a boost immunization and CD8^+^ T cell-stimulating epitopes to enhance protection over the longer term via the production of memory T cells. In addition, broadening the CD4^+^ T cell specificity reduces the total frequency of IFN-γ-positive CD4^+^ T cells but still improves the outcome of the vaccination. These studies demonstrate that the company that vaccine peptide constituents keep determines the harmful versus protective capacity of a vaccine. This has a direct clinical application since there are no efficient medicines in the clinic for chronic toxoplasmosis. This balanced approach to vaccine development confers robust protection against toxoplasmosis while avoiding serious immunological complications and preventing morbidity and mortality.

## Figures and Tables

**Figure 1 vaccines-11-01485-f001:**
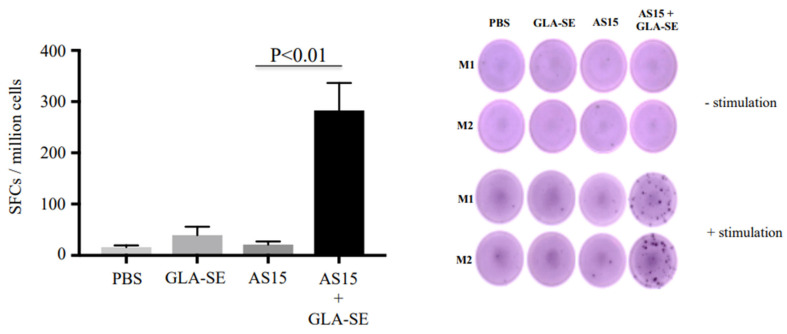
The *Toxoplasma* CD4^+^ T cell-stimulating peptide AS15 with the adjuvant GLA-SE (lipid TLR-4 agonist) results in increased induced IFN-γ. ELISpot showing IFN-γ spot formation from mouse splenocytes from untreated mice, those treated with GLA-SE or AS15, and those treated with AS15 + GLA-SE. Prior to measuring IFN-γ production, cells were stimulated with 20 µg/mL of AS15 peptide for 48 h. *p* < 0.05 for AS15 + GLA-SE IFN-γ ELISpots compared to controls. M1 (mouse 1) and M2 (mouse 2). Quantification of IFN-γ was evaluated in 2 separate experiments, Each experiment was carried out in triplicate. The data are from mice in both replicate experiments combined (*n* = 6).

**Figure 2 vaccines-11-01485-f002:**
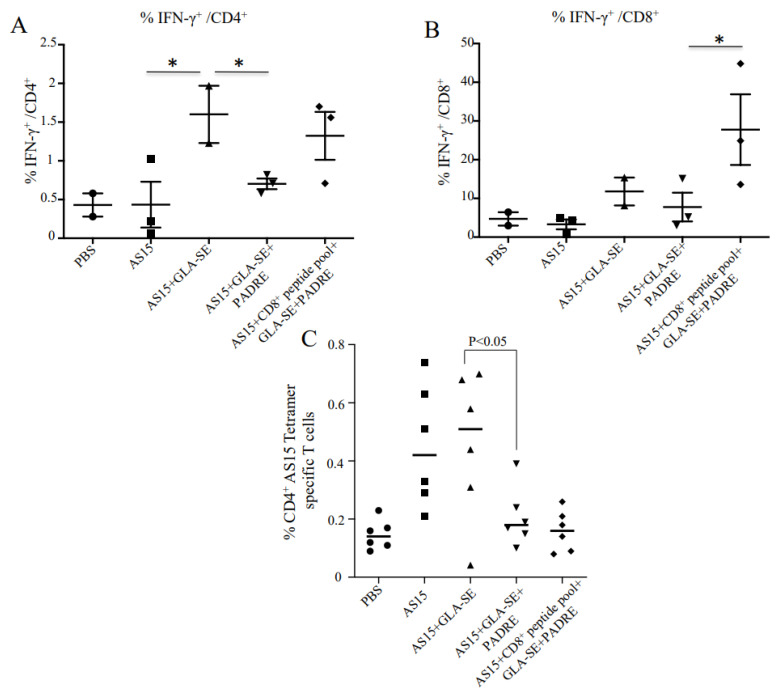
Immunization with adjuvanted AS15 plus and minus CD8^+^-stimulating HLA-A*11:01-restricted epitopes and the universal CD4^+^-stimulating epitope PADRE causes production of IFN-γ from CD4^+^ and CD8^+^ T cells, respectively. IFN-γ intracellular FACS analysis with peptide used as recall. IFN-γ production from CD4^+^ T cells (**A**) and CD8^+^ T cells (**B**) from mice vaccinated with AS15 peptide, AS15 + PADRE, AS15 + PADRE + GLA-SE, or AS15 + PADRE + GLA-SE + CD8^+^ T cell-restricted peptides was compared. Quantification of IFN-γ produced by mouse splenocytes was evaluated in 2 separate experiments. (**C**) Increased levels of CD4^+^ AS15 tetramer-specific T cells following stimulation indicates specificity of T cell response to AS15. The data in (**C**) are from mice in both replicate experiments combined (*n* = 6 mice). * = *p* < 0.05.

**Figure 3 vaccines-11-01485-f003:**
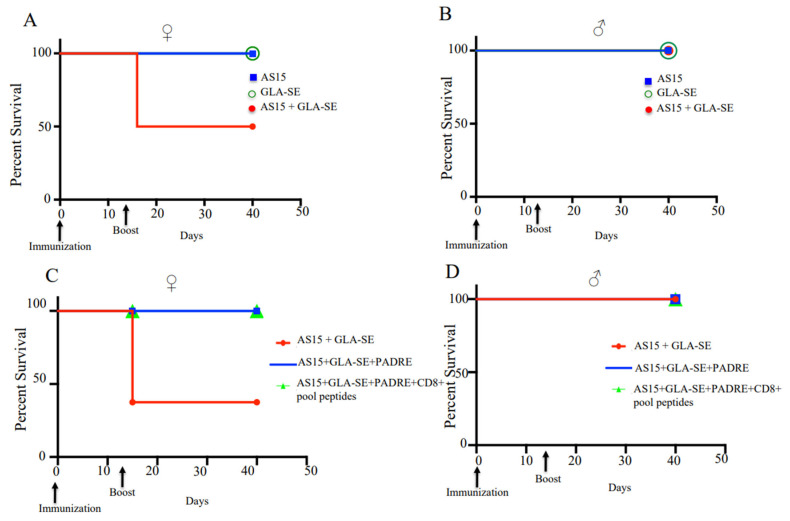
Boost immunization with adjuvanted AS15 results in the death of a subset of female HLA-A*11:01 transgenic mice, but is not lethal for males. Female and male uninfected mouse sensitivity to boosting with AS15 plus GLA-SE. At day 3–4 post-boost immunization, (**A**) 62% of female mice died, (**B**) while no deaths were observed in male mice. For each group of mice, (female *n* = 8 and male *n* = 15), differences were significant; *p* < 0.05, Student’s *t* test. (**C**) Addition of PADRE restores the survival of female mice. (**D**) No deaths were observed for both groups of male mice. This experiment is representative of 2 replicates.

**Figure 4 vaccines-11-01485-f004:**
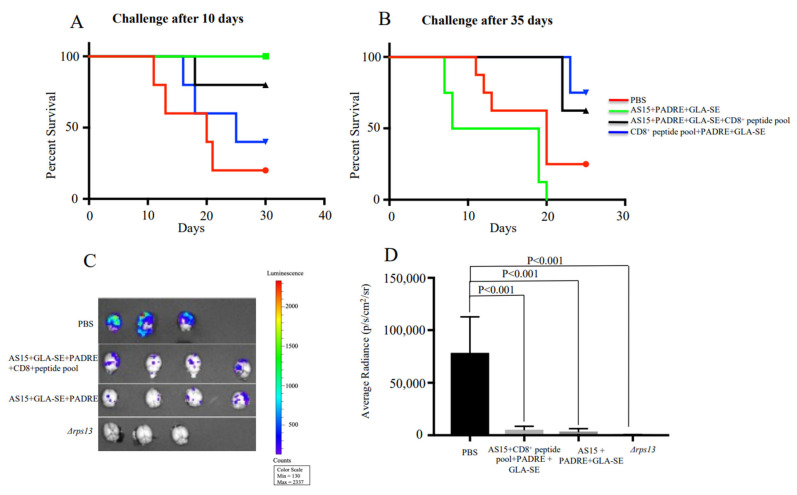
CD4^+^ T cell and CD8^+^ T cell response confers protection against challenge at early (10 days) and late (35 days) stage post-immunization, respectively. HLA-A*11:01 female transgenic mice were immunized with AS15 + PADRE + GLA-SE, AS15 + PADRE + GLA-SE + CD8^+^ T cell-restricted peptides or PADRE + GLA-SE + CD8^+^ T cell-restricted peptides 3 times at 2-week intervals. Then, 14 days after the last immunization, mice were infected with 2000 Me49 (Fluc) parasites at (**A**) 10 or (**B**) 35 days. Survival rates of mice as recorded and Kaplan–Meier curves were generated. The group of mice immunized with AS15 + GLA-SE was not included in this experiment. The data in the figure are from mice in both replicate experiments combined (*n* = 8 control and 8 immunized mice). (**C**,**D**) Number of luciferase-expressing parasites in the brain. *Δrps13* (conditional live attenuated *Toxoplasma* deficient in ribosomal small subunit protein 13) was used as a positive control.

**Figure 5 vaccines-11-01485-f005:**
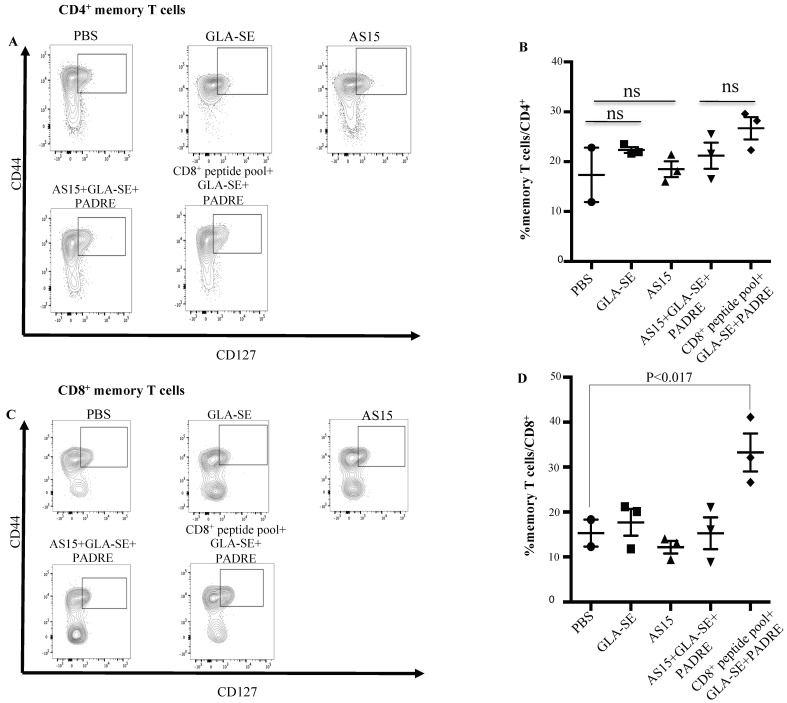
HLA-A*11:01-restricted peptides induce production of CD8^+^ memory T cells. Splenocytes from immunized mice were evaluated for CD8^+^ and CD4^+^ memory T cells. Memory T cells were defined as CD44^hi^CD127^+^. (**A**,**B**) CD4^+^ memory T cells. Flow cytometry gating for CD4^+^ memory T cells. Spleen cells are gated on CD3^+^CD4^+^ T cells. Memory T cells were defined as CD44hiCD127+. For each group, a representative FACS plot is shown with the percent of CD4^+^ memory T cells shown. (**C**,**D**) CD8^+^ memory T cells. Flow cytometry gating for CD8^+^ memory T cells. Memory T cells were defined as CD44^h^iCD127^+^. For each group, a representative FACS plot is shown with the percent of CD8^+^ memory T cells shown. ns: not statistically significant. The experiments were performed twice (*n* = 3 mice in each group).

**Figure 6 vaccines-11-01485-f006:**
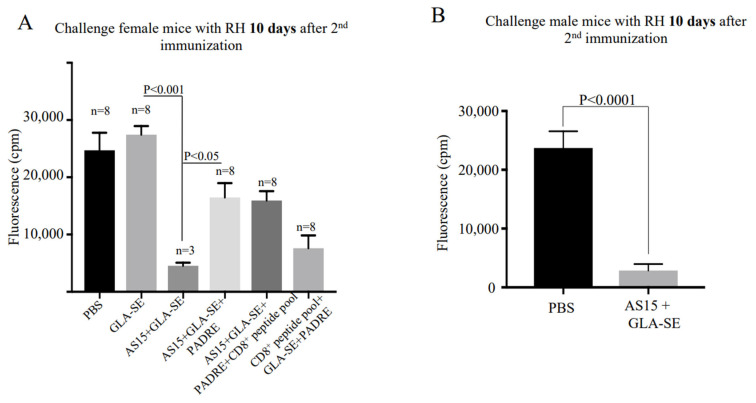
Immunization confers reduction in the virulent Type I *T. gondii* burden. Peritoneal fluid was collected 5 days post-infection from (**A**) female and (**B**) male mice with Type I parasites. YFP parasites were quantified using a fluorimeter. Differences between immunized mice before and after the second boost were significant (*p* < 0.01).

**Figure 7 vaccines-11-01485-f007:**
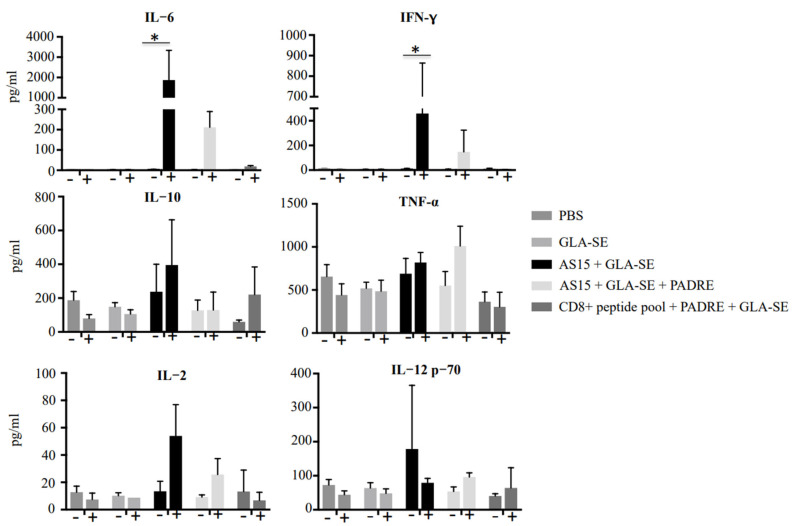
Immunization of HLA-A*11:01 transgenic mice with vaccine formulations containing GLA-SE-adjuvanted AS15 as well as CD8^+^ T cell-stimulating epitopes resulted in robust increases in circulating cytokines. Serum levels of IFN-γ, TNF-α, IL-12p70, IL-2, IL-10, and IL-6 measured with a cytokine bead array at 24 h before and after the second immunization. * = *p* < 0.05.

**Figure 8 vaccines-11-01485-f008:**
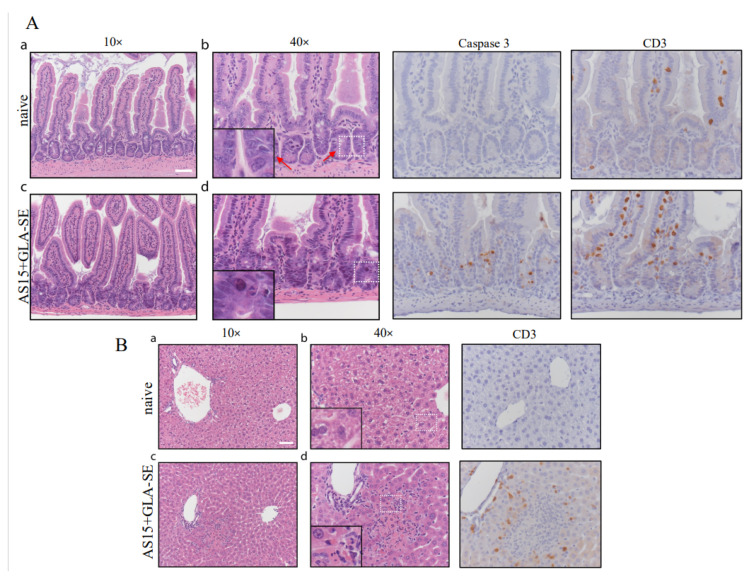
Immunization with adjuvanted AS15 alone results in severe intestinal and liver pathology in female mice. H&E micrographs of small intestine and liver from female mice at 24 h post-booster immunization with AS15 + PADRE. Small intestinal crypt apoptosis was increased in immunized mice and highlighted by caspase 3 staining. **A** (a–d) In the small intestine of immunized mice CD3 positive intraepithelial lymphocytes and apoptosis as confirmed by immunohistochemical staining for cleaved caspase 3 were more prominent that in control. **B** (a–d) Significant changes were also observed in the livers of immunized mice, which demonstrated microscopic necrotic foci with acidophil bodies, and polymophonuclear leukocytes, which were surrounding CD3^+^ T cells. Control naïve mice did not receive any immunogens. Scale bars represent 50 μm. Control naïve mice did not receive any immunogens.

**Figure 9 vaccines-11-01485-f009:**
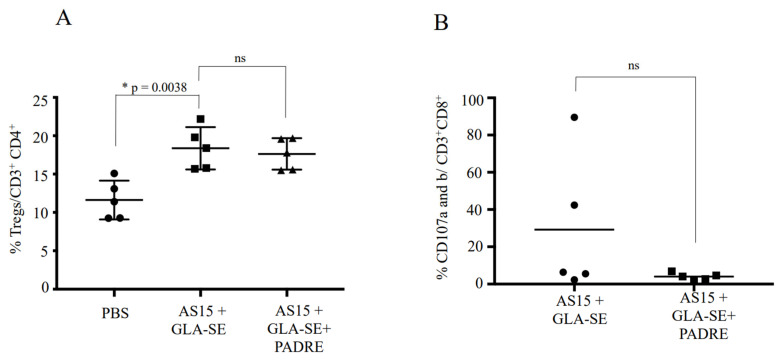
Inclusion of the universal CD4^+^ T cell-stimulating epitope PADRE did not lead to increased production of Tregs, but did result in lower levels of CD107a- and -b-expressing cells, suggesting a mechanism for abrogation of the severe phenotype in GLA-SE-adjuvanted AS15 immunized mice. The cytotoxic function of CD8^+^ T cells from female mice immunized with either AS15 + GLA-SE or AS15 + GLA-SE + PADRE is shown. The level of CD107 a/b expression on the surface of lymphocytes is used as a direct assay to quantify the cytotoxic T cells using CD107a and -b expression. Spleen cells from different female mice immunized with either AS15 + GLA-SE or AS15 + GLA-SE + PADRE were stimulated with AS15 peptide in the presence of FITC-conjugated anti-CD107a and -b and Golgi-Stop for 6 h. AS15 peptide was used at a final concentration of 20 μg/mL. (**A**) shows the means ± standard deviations of the percentages of Treg cells per CD4^+^ in the presence of AS15 peptide alone or in combination with PADRE and (**B**) shows the means ± standard deviations of the percentages of CD107a/b^+^ per CD8^+^ T cells in the presence of AS15 peptide. * statistically significant.

## Data Availability

Data sharing is not applicable to this article.

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
