# Peer review of "CD4+ T Cell Responses to Toxoplasma gondii Are a Double-Edged Sword"

_vaccines, 2023, doi:10.3390/vaccines11091485_

Round 1

Reviewer 1 Report

For the purpose of developing an appropriate human vaccine candidate against Toxoplasma gondii, the authors selected an optimal combination of adjuvants, CD4+ and CD8+ T-cell stimulatory protein, and a CD4+ T-cell epitope that can elicit an effective but not harmful immune response. Individual combinations were screened in laboratory mice on the basis of cytokine production, reduction of parasite load (tachyzoites) or number of brain cysts, histopathological changes and survival rate.

 A very thoughtful and elaborate study that may become an important step towards the development of a much-needed vaccine against Toxoplasma gondii.

Author Response

Response: We appreciate receiving the Reviewer’s thoughtful and laudatory comments. We have revised our manuscript and hope our manuscript now be found suitable for publication in Vaccines.

Reviewer 2 Report

The study is rather complex. However, it provides new information in the field. It should be interesting to investigate the cause of the adverse effects in female mice.

The main and ultimate question of this research is to develop a vaccine for humans able to confer protection against Toxoplasma gondii  but avoiding adverse effects. In particular, the effect of AS15 peptide immunization in conjunction with the adjuvant GLA-SE in HLA-A*11:01 transgenic mice was studied. 

The study addresses numerous aspects of the issue and can be, at some points, of not immediate understanding.  However,  in my opinion, this study provides new information in the field.

It highlights that the type of adjuvant and epitope may change the immune response, so that their choise need to be balanced. In particular, adverse effects were observed in a subset of boosted female mice; it should be interesting to investigate the cause of the adverse effects in female mice.

The study is rather complex and very articulated in the experimental methodological part. Some experiments were carried out with a low number of animals per group ( i.e. the Immunization with adjuvanted AS15 plus and minus CD8+ stimulating HLA-A*1101 re- 189 stricted epitopes and the universal CD4+ stimulating epitope PADRE, shown in Figure 2) and the statistical analysis may be partly affected by the small sample size; a Fisher test could be more appropriate. However, given the high number of experiments, I understand that it may be difficult to set up an appropriate set of in vivo tests. For this reason, I raised no objections.

Conclusions are consistent with the results showed in the study and also raise some questions about the effects of combinations of epitopes and adiuvants and their safety in specific group of animals

Most of the Figures are clear. Only figure 3 and figure 4 (A and B) were more difficult to understand, but this may be my limit.

Reviewer 3 Report

1. In Figure 2C, it seems that there was no significant difference of the %CD4+ Tetramer+ cells in each group. In your text, you got the conclusion that AS515 significantly increase the CD4+tetramer specific T cells and PADRE significantly diminished the CD4+tetramer specific T cells. Can you add more mice or redo the statistical analysis?

2. Could you please explain why female mice but not male mice boosted with AS15+GLA-SE have enhanced mortality? Is IL-6 cytokines storm associated with estrogen? 

3.  Could you explain why challenge time (10 days and 35 days Figure 4 A and B) could change a lot of the survival of mice especially after immunizing with AS15+PADRE+GLA-SE but not other combines? What gender of mice do you use? Is there any difference of survival of female and male mice with immunization with those combines? 

4. According to your results, is there any clues indicate that AS15+PADRE+GLA-SE is promising to apply in human beings to protect us from Toxoplasma gondii? It there any potential difference of applying Toxoplasma gondii vaccines to mice and human?

Author Response

Comments and Suggestions for Authors

Response: We appreciate the Reviewer’s positive comments and have modified the construction of the manuscript as outlined with the specific comments.‎ 

Comment 1: In Figure 2C, it seems that there was no significant difference of the %CD4+ Tetramer+ cells in each group. In your text, you got the conclusion that AS515 significantly increase the CD4+tetramer specific T cells and PADRE significantly diminished the CD4+tetramer specific T cells. Can you add more mice or redo the statistical analysis?

Response: As suggested, we combined the mice from the two replicate experiments and modified the statistical analysis. We added this section to the manuscript on page 6. “Increased levels of CD4+ tetramer specific T cells following stimulation indicate the specificity of T-cell response to AS15. When CD8+ T cells eliciting peptide was added in the immunogens, % tetramer positive CD4+ T cells diminished. The mean of tetramer-positive CD4+ T cells also significantly diminished when PADRE was added (p<0.05), suggesting changes in the distribution of CD4+ T cells when other immunogens are added.” In addition, we corrected the legend in Figure 2C and added the following sentence “The data in Figure 2C are from mice in both of 2 replicate experiments combined (n= 6 mice).”

Comment 2:  Could you please explain why female mice but not male mice boosted with AS15+GLA-SE have enhanced mortality? Is IL-6 cytokines storm associated with estrogen? 

Response: An interesting finding in the design of our vaccine is that upon secondary immunization with the AS15 peptide and the TLR-based adjuvant, female but not male mice experienced significant and lethal immunopathology, which appeared to be modulated by incorporation of another CD4 helper epitope. This is important to find for the future vaccine design to prevent toxoplasmosis. Using many technical tools (ELISA, FACS, ELISpot, intestinal and liver imaging), we saw a significant increase in the level of IFN-γ and IL-6, and most importantly, immune suppressive cytokines like IL-10 which suggests some suppressive mechanism at play by the PADRE-specific CD4+ T cells, which is critical to know going forward in vaccine design. The phenotype changes were carefully investigated in female mice, and it will be of interest in the future to determine whether these cytokines changes happen in males. We agree with the reviewer’s suggestion that IL6 storm observed in females immunized with AS15 + PADRE should be investigated in male mice and whether there is an association with estrogen.

3. Could you explain why challenge time (10 days and 35 days Figure 4 A and B) could change a lot of the survival of mice especially after immunizing with AS15+PADRE+GLA-SE but no other combines? What gender of mice do you use? Is there any difference of survival of female and male mice with immunization with those combines? 

Response: To make sure everything is clear for the reader, we added needed detail about the immunization schedule for these mice to figure 4A and B legends. The following sentence was added on page 24, line 551 “HLA-A*11:01 female transgenic mice were immunized with AS15+PADRE+GLA-SE, AS15+PADRE+GLA-SE+CD8+ T cells restricted peptides or PADRE+GLA-SE+CD8+ T cells restricted peptides 3 times at 2-week intervals. 14 days after the last immunization, mice were infected with 2,000 Me49 (Fluc) parasites at (A) 10 or (B) 35 days.” We also added this paragraph to the results section on page 11, line 249 “Taken all together, AS15 peptide immunization conferred protection only when a challenge occurs very close to the time of immunization (10 days), but provided no longer-term protection (35 days). Only CD8-stimulating peptide pools provided long-term protection, and the addition of AS15 was not helpful.” These experiments were performed in female mice, and future additional experiments for comparison with male mice will be of interest.”

4. According to your results, is there any clues indicate that AS15+PADRE+GLA-SE is promising to apply in human beings to protect us from Toxoplasma gondii? It there any potential difference of applying Toxoplasma gondii vaccines to mice and human?

Response: Our previous and current work lays the foundation for creating a novel vaccine to reduce or eliminate the initial T. gondii parasite burden. We used the best combination of immunogens, adjuvants, and deliveries necessary for a first-ever vaccine to prevent human toxoplasmosis. Using human HLA-11:01 transgenic mice, our results show that the adjuvants GLA-SE and the immuno-stimulating parasite CD4 and CD8 epitopes promote mouse survival to challenge. However, none of these vaccine regimens provided 100% complete protection. More peptides for sufficient population coverage will be needed to bring our immunogen prototype closer to clinical trials.

Reviewer 4 Report

The study explores the development of a vaccine for Toxoplasma gondii infection, for which no vaccines currently exist. Specifically, the researchers found that immunizing mice with the Toxoplasma CD4+ T cell-eliciting peptide, AS15, along with GLA-SE, offered protection against Type I and II Toxoplasma parasites. However, this combination also had the potential to cause a lethal cytokine storm, particularly in the intestines, liver, and spleen. The addition of PADRE decreased the side effect, while adding CD8+ restricted peptides provided further protection. The use of specific CD4+ T cell epitopes like AS15, along with PADRE and CD8+ peptides, demonstrates a promising direction, but one that requires careful calibration to prevent potential harm.

In general, this is an interesting study that provides further insights into developing vaccines for Toxoplasma gondii infection. However, some key data are missing in the paper. for example, there is no direct comparison of safety data between the AS15+GLA-SE group and the AS15+GLA-SE+PADRE+CD8+ peptide pool group. Additionally, there are several minor mistakes, such as the missing caption for Figure 3C/D. I recommend accepting this paper with major revisions. Check the details below.

Major concerns:

1.    In Fig2C, the authors should add a statistical test. Especially if they claim in the result session “Increased levels of CD4+ AS15 tetramer specific cells indicate T-cells responses are specific to AS15” that "The mean of tetramer-positive CD4+ T cells also significantly diminished when PADRE was added”, which doesn’t appear to match the figure.

2.    Fig 3C/D caption was missing.

3.    In Fig3C, authors compared mortality of boosted immunization of AS15+GLA-SE versus AS15+GLA-SE+PADRE groups. Ideally, they should include an extra group for AS15+GLA-SE+PADRE+CD8+ peptide pool to show that this group is also safer than AS15+GLA-SE group.

4.    In the result session (Fig4) “Peptide that elicits specific CD4+ T cells protects after immunization against challenge with Me49 parasites at early but not at late stages”, the author should include an extra group for AS15+GLA-SE condition. Without a direct comparison between the AS15+GLA-SE group and the AS15+GLA-SE+PADRE+CD8+ peptide pool group, it’s hard to conclude that stimulation of CD4+T cells by AS15 cannot provide a long-term protective effect, since the addition of PADRE diminished CD4+ T cells’ effect induced by AS15+GLA-SE.

5.    In the result session (Fig6A) “AS15 plus GLA-SE protects mice against virulent Type I Toxoplasma gondii”, authors should reduce the immunization dosage of AS15-GLA-SE group since 5/8 mice in this group died within 3 days after the second immunization.

Minor comments:

1.     When authors first mentioned the CD8+ T cells-stimulating peptides in the result session, “Addition of HLA A-11-restricted CD8+ T cell epitope peptides and/or PADRE modulates IFN-γ expression”, they should briefly introduce these peptides and include a citation.

2.     The result session (Fig5) “The addition of CD8+ T cells eliciting peptides plus GLA-SE to PADRE increases memory CD8+ T cell response” doesn’t seem to be directly relevant to sessions before or after that. Authors should explain why they did this experiment and what was the conclusion they made from it.

Author Response

Comments and Suggestions for Authors

The study explores the development of a vaccine for Toxoplasma gondii infection, for which no vaccines currently exist. Specifically, the researchers found that immunizing mice with the Toxoplasma CD4+ T cell-eliciting peptide, AS15, along with GLA-SE, offered protection against Type I and II Toxoplasma parasites. However, this combination also had the potential to cause a lethal cytokine storm, particularly in the intestines, liver, and spleen. The addition of PADRE decreased the side effect, while adding CD8+ restricted peptides provided further protection. The use of specific CD4+ T cell epitopes like AS15, along with PADRE and CD8+ peptides, demonstrates a promising direction, but one that requires careful calibration to prevent potential harm.

In general, this is an interesting study that provides further insights into developing vaccines for Toxoplasma gondii infection. However, some key data are missing in the paper. for example, there is no direct comparison of safety data between the AS15+GLA-SE group and the AS15+GLA-SE+PADRE+CD8+ peptide pool group. Additionally, there are several minor mistakes, such as the missing caption for Figure 3C/D. I recommend accepting this paper with major revisions. Check the details below.

Response: We appreciate the Reviewer’s gracious and positive comments. As suggested, we have modified the construction of the manuscript as outlined with the specific comments.

Comment 1:  In Fig2C, the authors should add a statistical test. Especially if they claim in the result session “Increased levels of CD4+ AS15 tetramer specific cells indicate T-cells responses are specific to AS15” that "The mean of tetramer-positive CD4+ T cells also significantly diminished when PADRE was added”, which doesn’t appear to match the figure.

Response: We agree that this is an important point to include. As suggested, we added the statistical analysis and updated the Figure 2C.

Comment 2:  Fig 3C/D caption was missing.

Response: Thank you for pointing out this, and we apologize. As suggested, we added the following sentences to the legends of Figure 3, page 24, line 546 “ (C) Addition of PADRE restores the survival of female mice, (D) No deaths were observed for both groups in male mice.”

Comment 3: In Fig3C, authors compared mortality of boosted immunization of AS15+GLA-SE versus AS15+GLA-SE+PADRE groups. Ideally, they should include an extra group for AS15+GLA-SE+PADRE+CD8peptide pool to show that this group is also safer than AS15+GLA-SE group.

Response: We agree with the reviewer’s comment. As suggested, we have now incorporated the group of AS15+GLA-SE+PADRE+CD8peptide pool in Figure 3C and D. The following paragraph was added on page 10, line 226: “Addition of PADRE alone or in combination with CD8peptide pool in the immunogen eliminated the increase in mortality (Figure 3C and D).

Comment 4:  In the result session (Fig4) “Peptide that elicits specific CD4+ T cells protects after immunization against challenge with Me49 parasites at early but not at late stages”, the author should include an extra group for AS15+GLA-SE condition. Without a direct comparison between the AS15+GLA-SE group and the AS15+GLA-SE+PADRE+CD8peptide pool group, it’s hard to conclude that stimulation of CD4+T cells by AS15 cannot provide a long-term protective effect, since the addition of PADRE diminished CD4+ T cells’ effect induced by AS15+GLA-SE.

Response: We appreciate the reviewer’s comment. To clarify, we did not include the group of AS15+GLA-SE in our subsequent survival analyses, as stated on page 6, line 219, because of the reproducible death of mice after the second immunization. Therefore, to further analyze the contribution of CD4+ T and CD8+ cells in the protection of mice after 3 times immunization at 2-week intervals, we exclude the AS15+GLA-SE group.

Comment 5: In the result session (Fig6A) “AS15 plus GLA-SE protects mice against virulent Type I Toxoplasma gondii”, authors should reduce the immunization dosage of AS15-GLA-SE group since 5/8 mice in this group died within 3 days after the second immunization.

Response: Excellent point, but this study wants to investigate the surprising and reproducible death of female mice in the AS15-GLA-SE group following a boost immunization. We agree with the reviewer’s comment that a further and a depth study on the analysis of different concentrations of AS15-GLA-SE should follow.

Comment 6:  When authors first mentioned the CD8+ T cells-stimulating peptides in the result session, “Addition of HLA A-11-restricted CD8+ T cell epitope peptides and/or PADRE modulates IFN-γ expression”, they should briefly introduce these peptides and include a citation.

Response: The CD8+ T cell epitope peptides list was described in detail in Materials and Methods on page 2, lines 94-96. We agree that such details should be incorporated in the result session. In addition, we have included the reference of El Bissati et al., JCI Insight (2016), and Scientific Reports (2020). 

Comment 7: The result session (Fig5) “The addition of CD8+ T cells eliciting peptides plus GLA-SE to PADRE increases memory CD8+ T cell response” doesn’t seem to be directly relevant to sessions before or after that. Authors should explain why they did this experiment and what was the conclusion they made from it.

Response: Our vaccine studies show that the GLASE+PADRE+CD8 T cell peptide pool vaccine appears the best in eliciting protection. The CD8+ T cell stimulating epitopes with CD4+ T cell help and GLA-SE adjuvant afforded the protection of mice challenged upwards of 30 days after immunization. This protection was accompanied by statistically significant increases in circulating memory T cells (Figure 5C-D). No significant CD4+ T cell memory response increase was observed in any group (Figure 5A-B).

We also added this paragraph to the discussion section, page 22, line 484, for the clarity of the manuscript. “Herein, our study touches on a very important issue regarding the balance between the protective versus pathogenic effects of a vaccine against human toxoplasmosis. Using human HLA-11:01 transgenic mice, the universal HLA-DR and mouse IA-b binding PADRE epitope, adjuvants GLA-SE, as well as previous knowledge regarding immuno-stimulating parasite CD4 and CD8 epitopes in mice, demonstrate specific combinations that promote mouse survival to challenge. Our quest to find the best combination of immunogens reveals the importance of CD4+ to confer protection soon after a boost immunization and CD8+ T cell stimulating epitopes to enhance protection over the longer term via the production of memory T cells. In addition, broadening the CD4+ T cell specificity reduces the total frequency of IFN-γ positive CD4+ T cells but still improves the outcome of the vaccination. These studies demonstrate that the company that vaccine peptide constituents keep determines the harmful versus protective capacity of a vaccine. This has a direct clinical application since there are no efficient medicines in the clinic for chronic toxoplasmosis. This balanced approach to vaccine development confers robust protection against toxoplasmosis while avoiding serious immunological complications and preventing morbidity and mortality.”

We greatly appreciate the Reviewers’ comments and suggestions. We believe that incorporating these suggestions has improved our manuscript. We also are grateful that the Reviewers and Editor are allowing us the opportunity to respond to these suggestions. We hope our manuscript might now be found suitable for publication in Vaccines. Please contact us if there is any other information we can provide.

Thank you very much for your consideration.  

Round 2

Reviewer 3 Report

Accept the revised version. 

Reviewer 4 Report

The authors have adequately addressed all the concerns and questions raised in the initial review. The evidence provided substantively supports the claims made in the paper, and the arguments are well-structured and coherent. Therefore, I believe the manuscript is ready for publication in its current format.